# Sphygmomanometer Dynamic Pressure Measurement Using a Condenser Microphone

**DOI:** 10.3390/s23198340

**Published:** 2023-10-09

**Authors:** Žan Tomazini, Gregor Geršak, Samo Beguš

**Affiliations:** Faculty of Electrical Engineering, University of Ljubljana, 1000 Ljubljana, Slovenia; gregor.gersak@fe.uni-lj.si (G.G.); samo.begus@fe.uni-lj.si (S.B.)

**Keywords:** RF microphone, blood pressure, low frequency, oscillometric, BP simulator

## Abstract

There is a worldwide need to improve blood pressure (BP) measurement error in order to correctly diagnose hypertension. Cardiovascular diseases cause 17.9 million deaths annually and are a substantial monetary strain on healthcare. The current measurement uncertainty of 3 mmHg should be improved upon. Dynamic pressure measurement standards are lacking or non-existing. In this study we propose a novel method of measuring air pressure inside the sphygmomanometer tubing during BP measurement using a condenser microphone. We designed, built, and tested a system that uses a radiofrequency (RF) modulation method to convert changes in capacitance of a condenser microphone into pressure signals. We tested the RF microphone with a low-frequency (LF) sound source, BP simulator and using a piezoresistive pressure sensor as a reference. Necessary tests were conducted to assess the uncertainty budget of the system. The RF microphone prototype has a working frequency range from 0.5 Hz to 280 Hz in the pressure range from 0 to 300 mmHg. The total expanded uncertainty (k = 2, *p* = 95.5%) of the RF microphone was 4.32 mmHg. The proposed method could establish traceability of BP measuring devices to acoustic standards described in IEC 61094-2 and could also be used in forming dynamic BP standards.

## 1. Introduction

Cardiovascular diseases in humans are the leading cause of death globally, taking an estimated 17.9 million lives each year [1]. One of the main risk factors for cardiovascular disease is untreated hypertension, which can be diagnosed and monitored via blood pressure (BP) measurement [2,3]. It is critical to measure BP correctly as even a small measurement error can result in millions of wrong diagnoses of both hypertension and hypotension. Consequently, false treatments are being administered or no treatment at all when a treatment would be necessary [4,5].

Non-invasive BP measuring devices can determine systolic (SBP) and diastolic blood pressure (DBP) inside arteries indirectly by inflating a cuff that temporarily stops the blood flow inside the artery. In a pressure measuring device like a sphygmomanometer, correct detection of dynamic air pressure inside the tubes is important [6,7]. With static pressure, measurement forces are distributed equally in all directions. If the fluid is moving, another component called dynamic pressure is introduced in the direction of the flow. Currently, not enough importance is placed on this dynamic pressure component. Human BP can be derived from listening to Korotkoff sounds which occur when a blood pressure cuff changes the flow of blood through the artery [8,9,10]. This can be performed auscultatory by a trained medical professional using a stethoscope and an aneroid or mercury sphygmomanometer. An automated method for measuring blood pressure is performed with oscillometric BP monitors which replace the need for a human component by using a variety of algorithms to estimate SBP and DBP from the oscillometric pressure pulses occurring in the cuff pressure [6,7].

The main goal of this study is to design, implement and asses a novel measuring system for recording changes in air pressure inside the tubes of a sphygmomanometer during BP measurement using a condenser microphone. The capacitance of a condenser microphone changes with movement in its membrane, thus forming a capacitor with an immovable backplate electrode. Commonly, a constant charge is applied to the moving membrane, and change in capacitance alters the voltage across the microphone [11,12,13]. To the best of our knowledge the method of measuring air pressure inside the sphygmomanometer tubing with a microphone is a novel idea. This is due to the fact, that the majority of microphones are not built to withstand high static pressures (i.e., of 300 mmHg) that occur with PB measurements [14,15]. Most microphone capsules are also not designed for low frequency BP signals (i.e., 1 Hz) [16]. Low frequency measurements require a microphone with a well-controlled static pressure equalization with a very slow venting [17]. A condenser microphone with a preamplifier typically exhibits high-pass filter characteristic. To address this, we employed a radiofrequency (RF) method of converting pressure into signal. As shown by Urbansky and Zolzer, this method enables a microphone to generate a response, even at the lowest signal frequencies, down to 0 Hz (DC) [18,19,20].

We aimed to determine the potential of the proposed method for dynamic pressure measurement. If successful, using this method, we could establish the traceability of BP devices through calibration according to acoustic standards such as the primary method for pressure calibration of laboratory standard microphones by the reciprocity technique (IEC 61094-2:2009/AMD1:2022) [21,22,23].

## 2. Materials and Methods

### 2.1. Requirements of the Designed System

The measuring range, accuracy and resolution of our measuring system was chosen according to the characteristics of human BP. The pressure range was, therefore, from 0 mmHg to 300 mmHg, which is roughly 40 kPa or 186 dB SPL, respectively [24,25,26]. The same upper limit is commonly used in commercial automatic BP monitors. The frequency range we focused on is from 0.5 Hz to 5 Hz, which translates to a range from 30 BPM to 300 BPM, covering the range of the human heart rate [27,28]. However, we wanted to include higher harmonics and other heart noises, so the actual frequency range was up to 280 Hz.

According to ISO 81060-1 and ISO/IEC 80601-2-30, the maximum permissible error of a sphygmomanometer in use is set to ±3 mmHg for new and ±4 mmHg for devices already in use for at least a year [29,30]. In fact, the measurement error increases with aging [31]. We planned to assess if our system could achieve these specifications and if it had potential to be used as a BP device calibrator. As a reference, we used the GE Druck (UK, Leicester) pressure sensing platform UNIK-5000 for frequencies up to 10 Hz, and a GRAS (Denmark, Holte) 40BH microphone with a Brüel & Kjær (B&K) (Denmark, Nærum) 2669 preamplifier for frequencies above 10 Hz. For acoustic calibration reference, we used a sound calibrator B&K (Denmark, Nærum) 4231 which can produce 1 kHz tone at 114 dB SPL.

The main consideration when designing a BP measuring system using a microphone was the microphone itself, which must withstand high pressures. We decided on a condenser microphone because of its versatility and ability to incorporate it into a RF circuit to achieve a sufficient response at low frequencies. With an RF capacitance converter circuit, we used a GRAS 40BH 1/4″ condenser microphone capsule with a dynamic range upper limit of 193 dB SPL.

### 2.2. RF Microphone Circuit

In a condenser microphone, a membrane with a constant charge moves according to sound pressure changes. This movement generates a small difference in capacitance which manifests as a voltage change [11,12,13]. That alone is not suitable for our purpose due to the high-pass filter response at lower frequencies. The RF method of detecting changes in capacitance of a condenser microphone fixes this problem.

The RF method is used in special low-noise microphones with improved low frequency response. Due to their low impedance, the microphones are also more resistant to the effects of relative humidity of the environment. With the RF method, the change in capacitance is not directly converted into a proportional change in voltage but is used to modulate a RF signal generated inside the microphone with an oscillator circuit. With demodulation of the modulated signal, a pressure signal can be obtained. Changing capacitance can influence the frequency, amplitude, or phase of an RF signal [18,19,20,32]. 

A condenser microphone membrane with an electrode (capacitor) can be used directly as a part of an oscillator circuit. The focus of our study was on a system based on RF phase modulation. The measured capacitance of the GRAS 40BH microphone (including the 1/4″ to 1/2″ adaptor) was around 10 pF. This meant we could expect a very low difference in capacitance at a sound pressure change of 1 Pa, e.g., at a sensitivity of 0.44 mV/Pa, the expected change in capacitance is just 0.22 aF. 

A simplified schematic of the RF circuit we used is depicted in Figure 1. We aimed for changes in pressure to modulate the phase of the RF signal. A crystal oscillator generates a clock signal of 25 MHz, which is increased in frequency by a clock synthesizer and then lowered again using frequency divider. Thus, we could achieve an adjustable RF frequency, 50% duty cycle and lower phase noise because the frequency divider includes a separate power supply from the RF signal. The divided signal is connected to the RLC circuit and frequency mixer. The RF signal frequency we needed to set was determined by the resonant RLC part of the circuit. The ferrite inductor coil we used had an inductance of 220 μH. The resulting resonant frequency was 3.393 MHz. If tuned to this resonant frequency, there should be no signal present at the output of the mixer. To achieve the maximum magnitude of the demodulated signal we needed to move off the resonance peak to either side for 90° phase shift. The RF frequency at which we achieved the best sensitivity was 2.825 MHz.

Two RF signals arrive at the mixer: one is used to drive the mixer circuit and the other is modulated by sound pressure changes which change the capacitance of the LC resonator part of the circuit. To avoid loading of the RLC circuit by the mixer, a buffer amplifier is implemented. The RF signals are combined by the mixer, and the result is a DC offset with a superimposed demodulated signal produced by the changes in capacitance caused by the pressure. The demodulated signal is amplified and fed to the ADC. To avoid external electrical noise and interference coupling the whole circuit is on a single PCB and in direct proximity to the microphone.

The demodulated signal AC component is in the μV range, while the DC offset is in a range from 3 V to 5 V, depending on the circuit temperature and output voltage of the power supply batteries. Finding a suitable op-amp was, therefore, a challenge as we needed a low voltage noise density also at low frequencies (in our case in the 1 Hz range). We decided on a LT6018 op-amp by Analog Devices with 1.2 nV/√Hz typical voltage noise. At 0.1 Hz, this value is 2.9 nV/√Hz, which we calculated to be low enough for sufficient SNR of the output signal. The crystal oscillator was ABM11 by ABRACON, while the clock synthesizer was Si5351A by Silicon Labs. The buffer consisted of a bipolar junction transistor BFR183 by Infineon due to its low input capacitance (1.1 pF). The RF mixer consisted of a balun transformer ATB3225 by TDK and a TS5A9411 analog switch by Texas Instruments. 

The circuit includes multiple low pass filters set at 15 kHz to ensure removal of higher frequency components. The amplified and filtered signal was sampled using a data acquisition device. We used a cDAQ-9174 by National Instruments (NI) with an NI-9250 ADC module and an NI-9260 for DAC module with a sample rate at 51.2 kS/s per channel. Data acquisition and processing was performed in the LabVIEW environment by NI on a personal computer. Post processing was performed in MATLAB by MathWorks.

Due to the temperature, vibrational and electromagnetic noise sensitivity of the circuit, a protective housing was essential to eliminate influence from the surroundings. We used a B&K UA-0035 adaptor for mounting a 1/4″ measurement microphone to a 1/2″ preamplifier. By using this adaptor, it was possible to utilize the AQ-0015 part from an artificial ear (B&K 4135) as a casing for our PCB with the RF circuit while minimizing additional parasitic capacitance. Figure 2 below shows a cross-section (a) and a photo (b) of the finished RF microphone. 

As shown in Figure 2a, the ground connection is realized via gold-plated spring connectors on the bottom side of the PCB (yellow). We refer to the RF capacitance conversion circuit with the GRAS 40BH microphone attached as the RF microphone. The 40BH microphone was connected to the electronic circuit via the B&K UA-0035 adaptor which was screwed onto the B&K AQ-0015 housing the RF circuit. This in turn provided sufficient force to hold the PCB in place. A better fixation of the PCB should be foreseen to avoid unintentional movement of the PCB inside adaptor. Even a slight position change can influence capacitance of the circuit and, therefore, influence the sensitivity. The circuit required a few different voltages for correct operation (±12 V, 3.3 V and 1.2 V). To ensure the lowest level of noise, the power at the 1.2 V and 3.3 V settings was provided by NiMH batteries. A lead-acid battery was used for supplying ±12 V power. The actual temperature of the PCB was determined with a NTC resistor, which enables temperature compensation to be implemented. Figure 3 contains photos of finished RF microphone circuit, housing, and microphone.

### 2.3. LF Sound Source Design

For the purpose of testing, a reliable and repeatable low-frequency (LF) sound source was required. To achieve an accurate reproduction of LF signals, sufficient membrane displacement and speaker power was needed. We used a 6RS140 speaker by FaitalPRO which was an 8-ohm 200 W speaker with a 6.5-inch diameter membrane. The speaker was attached to a wooden board in an airtight manner with a 1-inch hole in the center. Inside the hole, a DP-0776 calibration adaptor from a pistonphone was attached to ensure sufficient airtightness when inserting a sensor. By using this adaptor, we can fit any 1/4″ and 1/2″ microphone and also a UNIK-5000 reference sensor. The constructed LF sound source is depicted in Figure 4.

The speaker itself is raised from the board using spacers, to ensure the membrane does not hit the board and cause unwanted additional harmonic distortion (HD). We could achieve maximum usable SPL 141.5 dB SPL (1.87 mmHg) with a total harmonic distortion (THD) of 0.4% at 280 Hz. The lowest frequency this system could generate with sine signal THD less than 0.5% was 0.5 Hz at 139.8 dB SPL (1.47 mmHg). We measured the resonant frequency at 500 Hz, which is well above our frequency range and is thus not problematic. The LF sound source can produce only sound signals, i.e., dynamic pressure. Here, static pressure is at atmosphere level with dynamic pressure superimposed on top. During BP measurement static pressure rises to much higher levels (300 mmHg) but the dynamic pressure component is in the range produced by this LF sound source. 

We measured the frequency response of the LF sound source from 0.5 Hz to 280 Hz using three different measurement systems: a UNIK-5000, the RF microphone, and a B&K 2669 preamplifier with 40BH microphone. By using the same microphone with both the B&K 2669 preamplifier and RF capacitance converter, we eliminated the need to consider the 40BH microphone response as they are the same in both cases. To measure the LF sound source frequency response, we recorded a 50 s long measurement at each frequency in the specified frequency range. Every 5 s, we averaged the amplitude of the produced tone and converted it to dB SPL. We repeated the measurements with each of the three sensors. We ensured that the seal between the LF sound source and inserted sensor was airtight, as leakage would affect the frequency response.

### 2.4. Determining RF Microphone Temperature Response

The RF converter circuit sensitivity was predicted to be susceptible to temperature changes. The temperature influences each electrical component characteristic and causes mechanical deformation of the circuit and contacts. This movement could cause a slight change in capacitance and influence the resonant frequency and operating point of the circuit. Temperature compensation was, therefore, required. We tested the finished RF microphone inside a temperature climatic chamber HygroGen2 by Rotronic from 20 °C to 40 °C in 5-degree increments. We inserted the RF microphone into a calibrator producing 114 dB SPL at 1 kHz and then placed the still inserted RF microphone inside the chamber. We let the RF microphone temperature stabilize for 30 min at each temperature in the calibration chamber before the SPL measurement was performed. 

### 2.5. Testing the System Using a Blood Pressure Simulator

Even though the LF sound source was using a 200 W speaker in an enclosed system, it was still not powerful enough to generate the required pressure values. To test the RF microphone measuring system under actual high-pressure conditions, we needed to utilize a BP simulator, an electro-mechanical device for testing oscillometric BP monitors [33,34].

We used a SmartArm NIBP Simulator by Clinical Dynamics in conjunction with an Omron M6 (HEM-7001-E) oscillometric BP monitor providing necessary static air pressure inside the cuff and tubing. This ensured the essential repeatability of the measurements. The simulator was set to generate oscillometric air pulses corresponding to SBP/DBP 180/130, 150/100, 120/80, 100/70, and 80/50 mmHg. 

To obtain a reliable measurement, the BP monitor’s air pump needs to inflate the cuff to 20 mmHg above the expected simulated BP. The Omron M6 measurement range is 0 mmHg to 299 mmHg, which means the monitor could generate up to 320 mmHg of pressure inside the cuff and tubing during operation. Usage of most types of condenser microphones is, therefore, not possible, as the membrane would critically deform under pressure. In our study, the GRAS 40BH microphone, a microphone with intended usage for high level impulse-sound measurements such as explosions was used to monitor the simulated oscillometric output of the NIBP simulator.

We connected the reference pressure sensor (UNIK-5000) and RF microphone (using 40BH) via silicone tubing, to the Omron M6 BP monitor and its cuff using T-pieces (Figure 5). The entire set-up was positioned at the same height to avoid the hydrostatic-head correction. 

The air pump inside the BP monitor raises the static pressure inside air tubes and the inflatable cuff placed around the cylinder of the NIBP simulator. The static pressure triggers the simulator to generate oscillometric pressure pulses, i.e., dynamic changes in pressure, exactly as they would occur during real-life BP measurement. The reference pressure sensor and RF microphone simultaneously sample the air pressure inside the cuff. Prior to the measurement, the RF microphone was calibrated using the B&K 4231 sound calibrator generating 1 kHz tone at 114 dB SPL. The total recording time of the BP simulator was 40 s, which was enough for the whole process of cuff inflating and deflating.

### 2.6. Pressure Data Processing 

The RF microphone device was sampling and converting the voltage into pressure units in real time. However, the postprocessing of the data for the purpose of statistical analysis and uncertainty budget calculations, was not in real time. 

To obtain the LF sound source frequency response, the signal magnitude values from each sensor were compared. The 50 s measurements at each frequency were segmented into 5 s intervals, from which the mean value and standard deviation were calculated. We used a FIR filter on data acquired by the 40BH microphone to compensate for the pressure vent equalization. The FIR filter response was generated based on a comparison of the UNIK-5000 and RF microphone signal magnitude in the frequency range from 0.5 Hz to 10 Hz. We used the FIR filter both times—when using the 40BH microphone with the RF converter and for the 40BH microphone with the B&K 2669 preamplifier—to compensate for pressure vent equalization. We used a zero-phase digital filter function which did not introduce phase shift to the signal. The data acquired by UNIK-5000 did not undergo any additional conditioning.

We calculated a temperature compensation curve for the RF microphone from the measured PCB temperature. We used a 2nd degree polynomial fit on the sound pressure data at each temperature point. The obtained polynomial equation could then be utilized to calculate the needed SPL correction depending on the PCB temperature. We measured the calibrator sound pressure in 180 s segments at each temperature point and then calculated average values and standard deviation for each 10 s. By doing this, we also obtained the temperature coefficient uncertainty contribution. We used the calculated SPL corrections in every measurement that included the RF microphone.

For the BP simulator measurements, we imported the whole 40 s segment into Matlab. We compared the RF microphone and the UNIK-5000 data. The pressure data sampled by the RF microphone were firstly passed through a FIR filter to compensate for pressure vent equalization and then underwent temperature compensation. We also subtracted a linear approximation derived from the rising DC offset caused by unstable battery power supply voltage due to battery discharging. 

## 3. Results and Discussion

### 3.1. LF Source Frequency Response

The frequency response of the LF sound source was measured using different sensors (Figure 6). The response was not linear due to resonance of the speaker and non-linearities caused by the geometry of the membrane and speaker housing. The starting SPL was 135.1 dB at a sound frequency of 1 Hz. At a sound frequency of 10 Hz and lower, we observed attenuation in the 40BH measurement with the B&K 2669 preamplifier. This was expected due to a combination of the microphone and preamplifier responses. In contrast, the UNIK-5000 and RF microphone did not exhibit this attenuation, even though the RF circuit used the same 40BH microphone.

The responses of the RF microphone and 40BH utilizing a preamplifier began to differ at around 15 Hz or less but were almost identical in SPL above that threshold. This means the RF circuit did not introduce any linear distortion by itself, as the 40BH response was the same in both cases. 

Comparing the UNIK-5000 and RF microphone results, we observed a constant difference in air pressure of around 0.2 dB. The difference could be explained by the fact that the static pressure equalization vent located in the rear of 40BH microphone was not exposed to the same pressure on both sides of the membrane. The response does not decrease with decreasing frequency, as in the case when vent is exposed to the sound field. In fact, it increases with decreasing frequency, because the fraction of stiffness due to the reactive pressure in the internal cavity, becomes smaller as it is equalized through the vent. The low frequency sensitivity increase was smaller for microphones having a low fraction of air-stiffness [35,36,37]. 

We noticed a discrepancy when measuring with UNIK-5000 at sound frequencies of 100 Hz and above, which can be seen in Figure 5. Due to the geometry and size of UNIK-5000 pressure sensor, resonant sound frequencies occurred inside its housing. We can deduce that this signal amplification is caused by resonance as it continues increasing in magnitude at higher frequencies at a greater pace than when we measured the response with reference microphones. We calculated the fundamental resonant frequency of UNIK-5000 to be higher, at 1.1 kHz, but this still causes subharmonics at lower frequencies (100 Hz and above). Because of that, we used UNIK-5000 as a reference only in a frequency range from 0.5 to 15 Hz. For signal frequencies of 15 Hz and higher, we used a 40BH microphone with B&K 2669 preamplifier.

Figure 7 shows the frequency response of the RF microphone from 0.5 Hz to 280 Hz with added error bars representing the estimated measurement uncertainty. The uncertainty was ±0.125 dB, which corresponds to ±4.32 mmHg at a full scale of 300 mmHg.

We used temperature compensation and vent correction when calculating the frequency response. Our reference sound calibrator had an expanded uncertainty of 0.11 dB (*p* = 95%). The combined error could be further decreased if a reference calibrator with lower uncertainty were used or if we calibrated the 40BH microphone using a primary method. Furthermore, a low frequency calibrator, e.g., GRAS 42AE [38] would be a suitable alternative to our LF sound source as it can produce calibration signals in the range of 0.1 Hz to 251 Hz at up to 140 dB SPL.

### 3.2. RF Microphone Temperature Compensation Curve

The resulting graph with the temperature compensation curve for the RF microphone in dB SPL is shown in Figure 8.

The temperature of the actual circuit was found to be few degrees higher than the surroundings because of the power dissipation over small surface of the electronic circuitry. The overall stability of the SPL magnitude was less than 0.02 dB at each individual measurement. We used a second degree polynomial fit, which we then used in other measurements for temperature compensation. The measured pressure signal magnitude changed by 0.26 dB over the 20 °C to 40 °C temperature range.

### 3.3. BP Simulator Measurements

The final comparison was performed under BP measurement conditions by using a simulator with UNIK-5000 as a reference. The static pressure ranged from 0 to 250 mmHg with superimposed dynamic (oscillometric) pressure pulses. The pressure of interest is in the time interval after the cuff starts deflating, as BP is determined from oscillometric pressure pulses which are generated when blood starts flowing again and hits the walls of the artery. 

The air pump increased the air pressure in the cuff up to roughly 20 mmHg above the simulated/expected SBP. In the case of a simulated BP of 150/80 mmHg, that value would be 170 mmHg. In a real-life scenario, such high pressure occludes the blood flow through the artery completely. After deflating occurs, dynamic pressure fluctuations could be observed (Figure 9b). The highest oscillometric pressure pulses had amplitudes of 10 mmHg.

We observed a slight difference of 3 mmHg at the end of cuff inflation period at the 15 s mark (Figure 9a). The measured pressure is higher with the RF microphone at this point. This overshoot could manifest because of non-linearities in the microphone membrane at such high pressures. We observed this difference becoming larger when using an even higher pressure of 250 mmHg, which agreed with our hypothesis. Further experimenting is required to confirm this. At 5 s after the peak, this difference disappeared and both signals were almost identical, which indicates the RF microphone was working as intended, even under high static pressures. If we subtract the two pressure signals with UNIK-5000 being the reference, the error *E* is within the calculated uncertainty of 4.32 mmHg, even at its peak. The difference between the pressure signal sampled with the RF microphone and with UNIK-5000 in the 150/100 mmHg simulation measurement is depicted in Figure 10.

While performing the test we observed that the static pressure measured with the RF microphone was constantly rising at a rate of 1.7 mmHg/min, while that measured with UNIK-5000 did not exhibit this characteristic. After testing with a fixed linear power supply, we confirmed the static pressure drift was due to battery voltage decreasing during operation. This drift influence does not appear to affect the dynamic pressure measurement. We compensated for this trait in data processing by subtracting a linear function with a 1.7 mmHg/min slope. In the future, using a new ultra-low noise power supply design could eliminate this drift. 

### 3.4. Uncertainty Budget Estimation

Finally, we estimated the total uncertainty budget of the RF microphone. The uncertainty budget was calculated from measurements we conducted at a variety of dynamic and static pressures. We tested the system for factors that could influence the pressure measurement. The signal frequency contribution was determined by changing the frequency of the LF sound source and calculating the standard deviation at each measurement. The signal amplitude contribution was calculated after increasing the pressure signal generated by the LF sound source from 93.2 dB SPL up to its maximum of 141.5 dB SPL. To evaluate the reproducibility of the system, we measured a calibrator signal for a minute at a 51.2 kHz sampling rate. The same measurement was then repeated after dismantling and reassembling the system. The same position of the RF microphone inside the calibrator was ensured by using a stand. The uncertainty contribution caused by the temperature coefficient was estimated by calculating the maximum difference between the measured and fitted polynomial temperature curves in Figure 8. The reference calibrator uncertainty was taken from the sound calibrator certificate of B&K 4231. Calibrator positioning was measured by multiple reinsertions of the microphone in the calibrator. The comparison of the two measurements could then be used to calculate the calibrator positioning uncertainty. Repeatability tests were performed similarly, but without removing and inserting the microphone into the calibrator. Instead, we simply turned it off and on again over a span of 10 min.

The non-linearity contribution was estimated during the BP simulator measurement. We fitted a linear function to the difference *E* and found the biggest offset from the function value. To determine the effective resolution contribution, we measured the calibration signal for 5 min and searched for the maximum deviation. The hydrostatic head correction was calculated by assuming the maximum height difference between the sensors and equipment was 5 cm. 

The contributions to the combined uncertainty of the RF microphone are shown in Table 1. At 300.02 mmHg (40 kPa) the combined uncertainty equaled 4.32 mmHg. Since such high static pressures could not be generated, the data were extrapolated from the actual tests conducted at lower pressures. The temperature coefficient and calibration uncertainty were extrapolated from estimating 10 Pa sound pressure. The reproducibility, reference calibrator and repeatability were extrapolated from 31.7 Pa measurements. The contributions from signal frequency and resolution were measured at the highest possible usable pressure levels generated by the LF sound source, i.e., 23.6 Pa and 78.4 Pa, respectively. The signal amplitude was extrapolated from the pressure value of 0.92 Pa, as uncertainty was highest at low signal amplitudes. The non-linearity was calculated at 250.1 mmHg and extrapolated to 300.02 mmHg.

The 4.32 mmHg expanded uncertainty is too high to comply with ISO 81060-1 and ISO/IEC 80601-2-30 [29,30], which permits ±3 mmHg, but could be improved by calibrating the microphone by one of the primary methods and foregoing the need for any additional calibration. The reproducibility contribution is high because we were using battery-powered supply for multiple voltage levels. The power was constantly decreasing which impacted the measurements, especially the static pressure component. In the future, we will reduce this error source by using a fixed ultra-low noise power supply. 

The high uncertainty contribution was also due to the positioning of the RF microphone during calibration. A slight change in positioning would result in a large shift in calibration value. This should be investigated further, and a mechanism should be developed to physically fix the microphone in the same place during the calibration process. Our current set-up (acoustic calibrator) relies only on an airtight seal to hold the microphone in place. 

There are also other factors which should be tested and included in the total uncertainty budget of the RF microphone. These include the tube length, tube rigidity, total air volume inside the tubes, impact of atmospheric pressure, environmental temperature, and relative humidity. We assume the total contribution of the listed factors is rather negligible, like hydrostatic head-correction, which was calculated to be ±0.003 mmHg for a 5 cm height difference. Nevertheless, further testing should be conducted to confirm this.

## 4. Conclusions

In this paper, a novel method of measuring air pressure inside the cuff and tubing during BP measurement is proposed. We designed, developed, and tested a RF microphone system using a condenser microphone 40BH which could be used to measure oscillometric pressure pulses. 

RF microphones have sometimes been used for high pressure measurements in recent years [15], but never for the purpose of blood pressure measurement. There are published studies of BP measurement which implement a microphone [39], but the microphone was never directly positioned inside the cuff. From the results of our study, we conclude that the proposed method could be used as a BP measurement system and that the RF microphone could be a viable alternative to common calibration techniques using classical pressure sensors. With some improvements, it would be possible achieve the recommendations of standards ISO 81060-1 and ISO/IEC 80601-2-30 and use the RF microphone in sphygmomanometer calibration procedures. In addition, traceability to acoustic standards such as IEC 61094-2 is possible. The method is especially interesting in relation to dynamic pressure measurement, which seems to be lacking in current sphygmomanometer and automatic BP meter calibrations. As opposed to commonly used pressure sensors, condenser microphones are designed to detect dynamic pressure changes. The proposed method could, therefore, be used for setting dynamic pressure BP standards. In future studies, a comparison of the common piezoresistive pressure sensors used in commercial BP measuring devices and RF microphones should be conducted.

Using a low frequency calibrator, such as GRAS 42AE, as a sound source would also enable an assessment of the set-up in the frequency range between 0.1 Hz and 1 Hz signal at 140 dB SPL. Our LF sound source could only produce a usable sine signal with THD 0.5% down to 0.5 Hz at 139.8 dB SPL. The main reference we used was the UNIK-5000 piezoresistive pressure sensor. Due to the sensor housing, acoustic resonance started interfering with pressure signal much earlier than expected, at around 100 Hz sound frequency. This occurred even though the fundamental resonant frequency is above 1 kHz. We were planning to apply UNIK-5000 as a comparison with the RF microphone for the whole frequency range from 0.5 Hz to 280 Hz, but this was not possible because of this resonance. In future work it would be prudent to use a pressure sensor with a physically smaller casing as a reference to avoid resonant frequencies in the operating range.

The calculated RF microphone uncertainty of 4.23 mmHg could be improved by using more accurate reference, more stable power supply and better mechanical fixation during calibration. We envisage using a primary method for pressure calibration of laboratory standard microphones by the reciprocity technique described in IEC 61094-2 on the utilized condenser microphone. The uncertainty achieved with primary pressure reciprocity calibration was 0.04 dB (k = 2) at 32 Hz frequency in [40]. By using the primary method, we could minimize calibrator positioning and reference calibrator uncertainty in the uncertainty budget. Doing this, we could achieve the 3 mmHg expanded uncertainty needed for performing BP calibrations.

Additional changes to the RF circuit are planned. We tested various modulation techniques and finally decided on phase modulation of the RF signal. This could be repeated using amplitude or frequency modulation. We used different values of inductors in the RLC circuit, part of which was the condenser microphone. The SNR of the signal is directly dependent on inductor properties like *Q* factor and *L*. Currently, the RF circuit is based on a GRAS 40BH microphone and, therefore, incorporates its sensitivity and capacitance. Other condenser microphones capable of withstanding 300 mmHg pressures are planned to be used in the RF circuit.

## Figures and Tables

**Figure 1 sensors-23-08340-f001:**
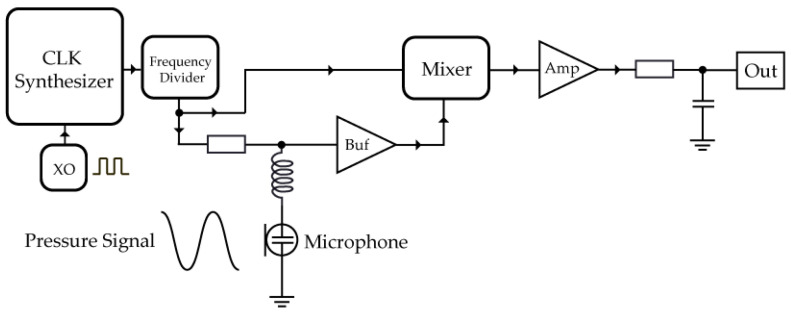
Schematics of the RF electronic circuit responsible for converting changes in pressure into voltage changes. The clock synthesizer generates an RF signal using a crystal oscillator (XO) which is then modulated by the pressure signal, changing the microphone capacitance. “Buf” represents a signal buffer and “Amp” a signal amplifier.

**Figure 2 sensors-23-08340-f002:**
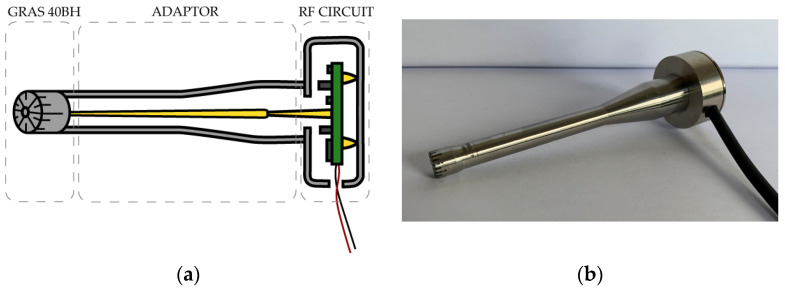
RF microphone measurement system cross-section diagram (**a**) and a photo of a finished prototype (**b**). The GRAS 40BH condenser microphone is connected to the RF circuit inside a metal casing. The adaptor is necessary for converting the 1/4″ thread to a 1/2″ one.

**Figure 3 sensors-23-08340-f003:**
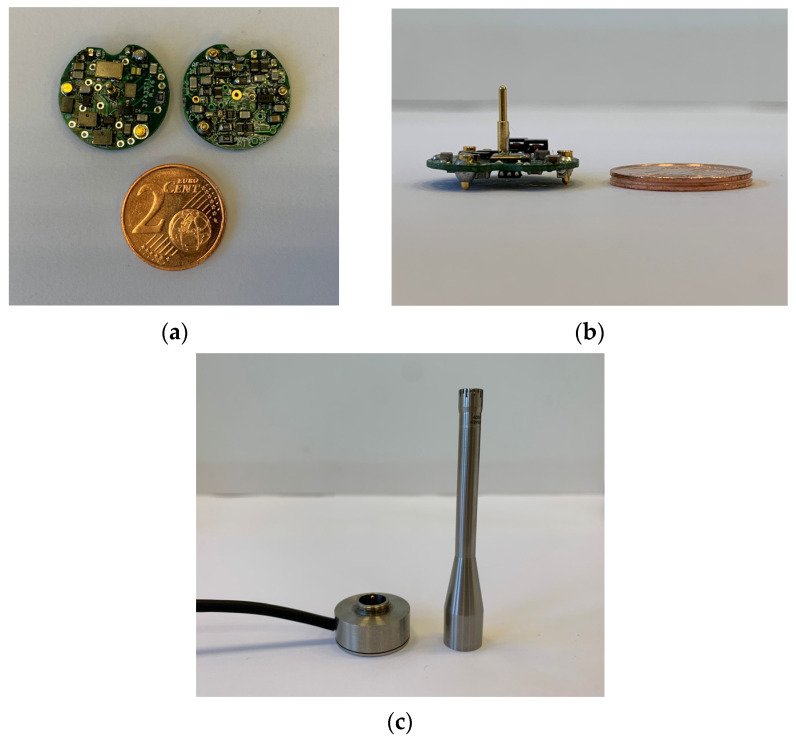
Top, bottom (**a**) and side view (**b**) of a prototype RF circuit. There is a 2-cent coin for size comparison. The gold-plated spring connector on top attaches to the microphone and the bottom connectors touch the B&K AQ-0015 metal casing. The RF circuit screwed inside the AQ-0015 can be seen on the left side in (**c**). On the right side in (**c**) is the B&K UA-0035 adaptor with the GRAS 40BH condenser microphone attached on top.

**Figure 4 sensors-23-08340-f004:**
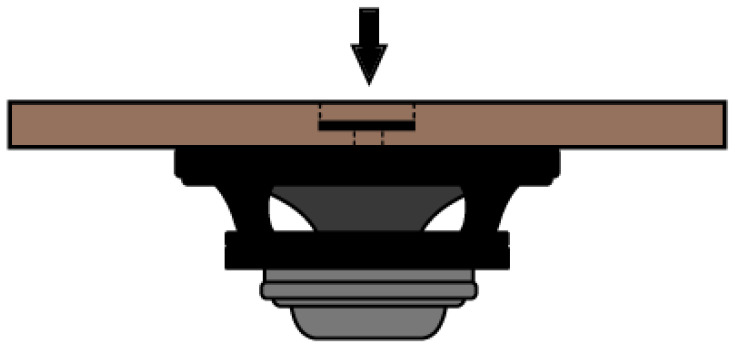
Side view of an LF sound source comprised of a FaitalPRO 6RS140 speaker (black and grey) attached to a heavy wooden board (brown). The center hole was sufficient for 1/4″ and 1/2″ microphones pistonphone adaptor. The RF microphone and other sensors could be inserted into the adaptor (arrow).

**Figure 5 sensors-23-08340-f005:**
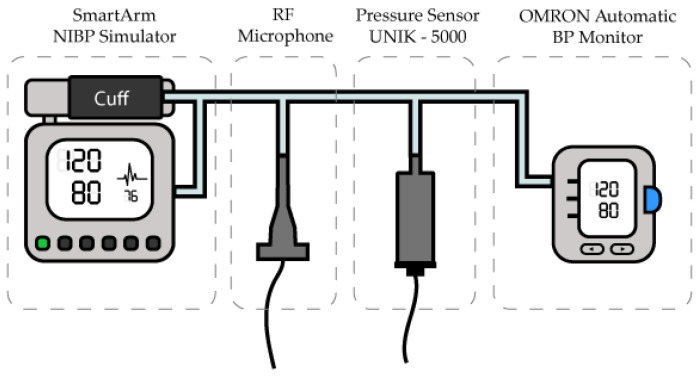
Measuring set-up using NIBP Simulator as the pressure generator. The Omron M6 BP monitor provided the static pressure needed for simulator operation. The two sensors are connected to the tubing in the middle.

**Figure 6 sensors-23-08340-f006:**
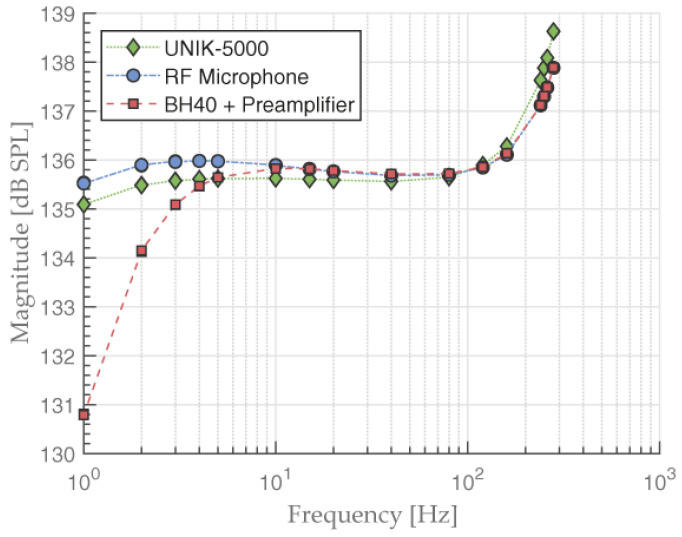
Frequency response of LF sound source measured from 1 to 280 Hz by a UNIK-5000 pressure sensor, a GRAS 40BH microphone with an RF capacitance converter and a GRAS 40BH microphone using a BK-2669 preamplifier.

**Figure 7 sensors-23-08340-f007:**
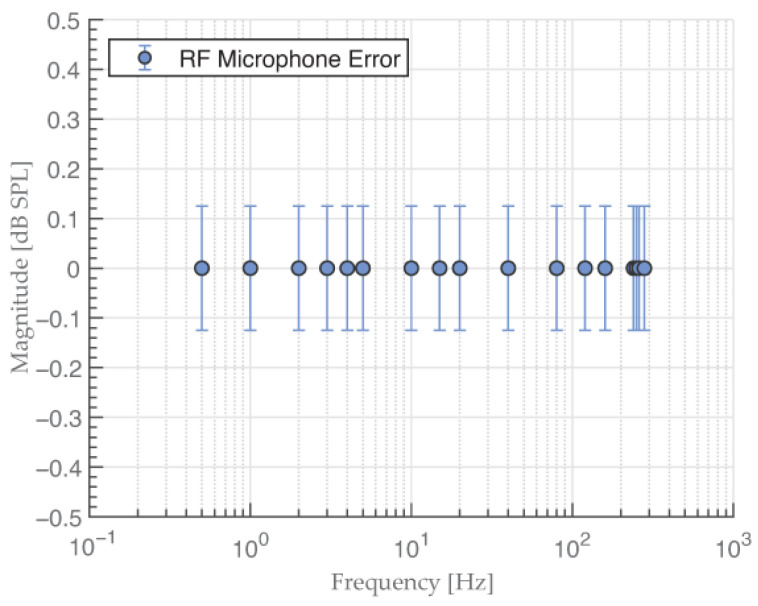
RF microphone frequency response with added uncertainty range (error bars). The uncertainty was within ±0.125 dB at all measured frequencies.

**Figure 8 sensors-23-08340-f008:**
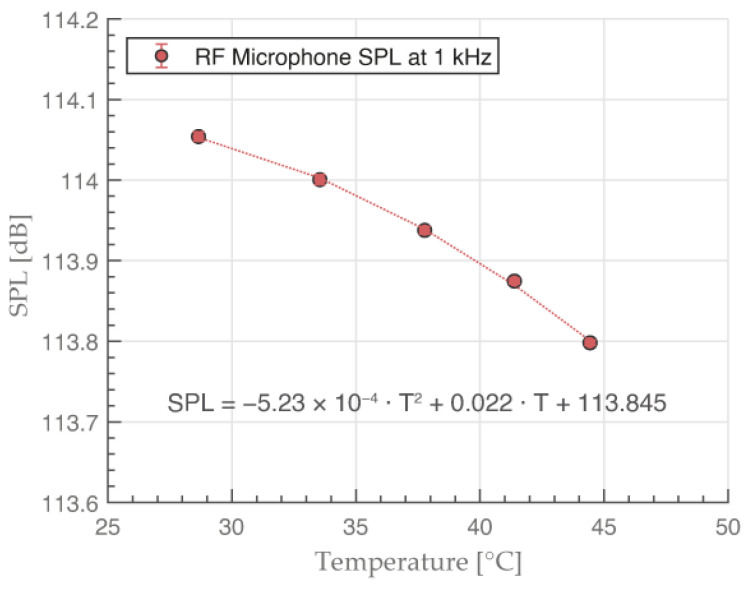
RF microphone temperature compensation curve in the 20 °C to 40 °C temperature range. The sound source generated a 114.05 dB SPL tone at 1 kHz frequency.

**Figure 9 sensors-23-08340-f009:**
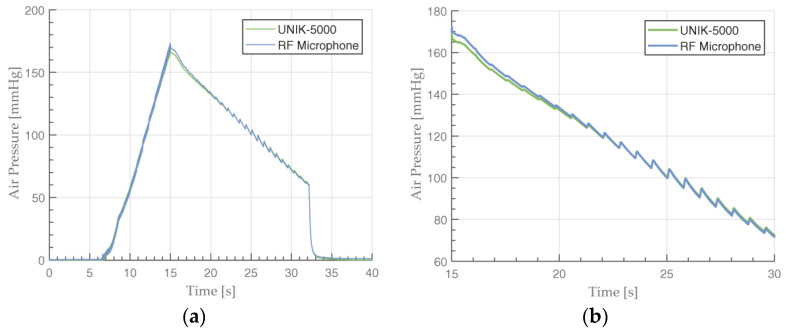
Comparison of a BP simulator pressure measurement performed by UNIK-5000 pressure sensor and RF microphone (**a**) and 15 s close-up (**b**). The simulation profile used is 150/100 mmHg with a 117 bpm heart rate.

**Figure 10 sensors-23-08340-f010:**
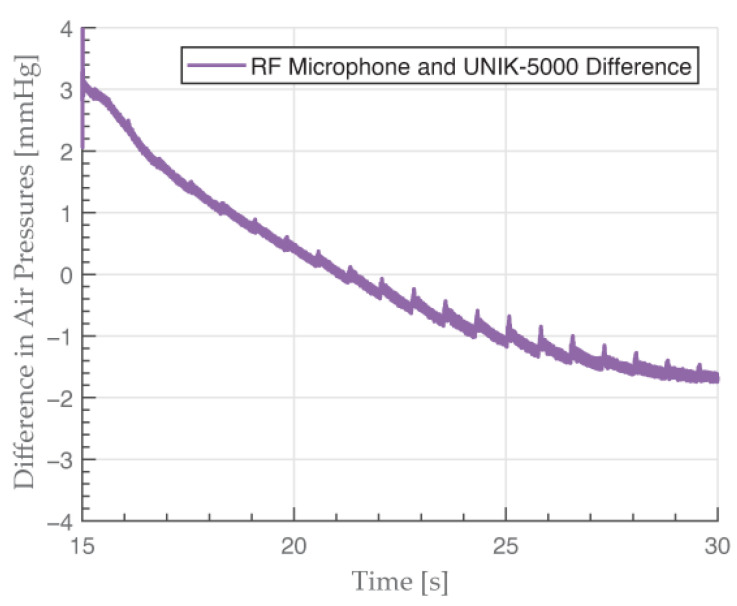
Difference between pressure signal sampled with the RF microphone and UNIK-5000. The error is within the expanded uncertainty range of 4.32 mmHg.

**Table 1 sensors-23-08340-t001:** Measurement uncertainty budget of RF microphone at 300 mmHg. The expanded combined uncertainty was 4.32 mmHg, with reference calibrator and reproducibility being the biggest contribution to the total uncertainty.

Quantity X_i_	Estimate x_i_ (mmHg)	Standard Uncertainty u(x_i_) (mmHg)	Probability Distribution	Sensitivity Coefficient c_i_	Uncertainty Contribution u_i_(y) = c_i_u(x_i_) (mmHg)
Signal frequency	300.02	0.67	square	1.00	0.67
Signal amplitude	300.02	0.21	square	1.00	0.21
Reproducibility	300.02	1.16	normal	1.00	1.16
Temperature coefficient	300.02	0.14	square	1.00	0.14
Reference calibrator uncertainty	300.02	1.01	normal	1.00	1.01
Calibrator positioning	300.02	0.94	square	1.00	0.94
Repeatability	300.02	0.18	square	1.00	0.18
Non-linearity	300.02	0.67	square	1.00	0.67
Resolution	300.02	0.63	square	1.00	0.63
Hydrostatic head correction	300.02	0.00	square	1.00	0.00
Combined uncertainty	-	-	-	-	2.16
Expanded uncertainty (k = 2, *p* = 95.5%)	-	-	-	-	4.32

## Data Availability

The data that support the findings of this study are available from the corresponding author upon reasonable request.

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
