# Peer review of "Sphygmomanometer Dynamic Pressure Measurement Using a Condenser Microphone"

_sensors, 2023, doi:10.3390/s23198340_

Round 1
Reviewer 1 Report
You used a novel method of measuring air pressure inside the sphygmomanometer tubing during BP measurement using a condenser microphone.I think it is original. However as you say in line 425,the 4.32 mmHg expanded uncertainty is too high to comply with ISO 81060-1 and ISO/IEC 80601-2-30,so I think the sensors need improvement beforepaper published,.Besides, there are some other problems with the paper.
1. In line 212, the temperature in test is from 20 °C to 40 °C . However, during the actual blood pressure monitoring,the ambient temperature range is wider. For example, in winter, the temperature will be lower than 20 degrees. So the test temperature is not enough.
2. In line 226, your test has different SBP/DBP. However your Chapter 3.3 only show the result of 150/100.I think it is not enough, such as the max differences in other different SBP/DBP should be shown also.
3. In my opinion, the test of actual humen blood pressure measurement maybe should be experimentized to further prove the reliability of the sensor.
Reviewer 2 Report
The authors proposed a novel method of measuring air pressure inside the sphygmomanometer tubing during BP measurement using a condenser microphone. This is an important amazing and creative work as far as I am concerned. Therefore, I believe this manuscript can be accepted after a minor revision.
1. More physical image regarding the device should be provided in the manuscript.
2. The article can be written more succinctly, please refer to and cite DOI: 10.1038/s41467-022-33454-y
The quality of English Language is good.
Reviewer 3 Report
The problem is well presented and the solution proposed is convincing.
However, for a journal article, certain points should be improved before publication. Among the most important:
- give the references of the components used. Some are given and others not...
- tests with better quality power supplies should be carried out. This cannot remain in perspective.
- we don't really understand if the signal processing is in real time or not. This type of tool must provide information in real time!
- tests are carried out on a simulator, but none on people even though this is non-invasive. This raises questions.
- some columns in table 1 seem unnecessary.
No comment
Reviewer 4 Report
This paper reports on the design, fabrication, and testing of a system that uses radio frequency (RF) modulation methods to convert changes in capacitance of a capacitor microphone into a pressure signal. The authors tested this RF microphone against a low-frequency sound source, a BP simulator, and a piezoresistive pressure sensor. The necessary tests were performed to evaluate the uncertainty budget of the system. The proposed method can be used to establish the traceability of blood pressure measurement devices to the acoustic standards described in IEC 61094-2 and can be used to form dynamic BP standards.
It is written on lines 57-59 that To counter the high-pass filter characteristic of a microphone with a preamplifier, we used a radiofrequency (RF) method that as shown by Urbansky and Zolzer provides a response down to DC signals. This statement is difficult for me to understand because radiofrequency and DC signals are relative concepts. A more detailed explanation is needed.
It is written on lines 264-265 that you calculated a temperature compensation curve for RF microphone from measured PCB temperature. I would like you to show us the data.
The LF sound source you used is shown in Figure 3. However, this figure is not cited in your text. Please cite Figure 3 appropriately.
Round 2
Reviewer 1 Report
The paper used a novel method of measuring air pressure inside the sphygmomanometer tubing during BP measurement using a condenser microphone. I think it is original. What’s more, the questions that I asked in last comments have been answered by authors and the paper has been revised. I think the paper can been accepted in present form.